# A Case of Atypical Unilateral Optic Neuritis Following BNT162b2 mRNA COVID-19 Vaccination

**DOI:** 10.3390/vaccines10101574

**Published:** 2022-09-20

**Authors:** Shuntaro Motegi, Takayuki Kanda, Masaru Takeuchi

**Affiliations:** Department of Ophthalmology, National Defense Medical College, 3-2 Namiki, Tokorozawa 359-8513, Japan

**Keywords:** COVID-19, optic neuritis, mRNA vaccine, vaccination

## Abstract

Background: We report a case of atypical unilateral optic neuritis after receiving the BNT162b2 mRNA-based COVID-19 vaccine. Case Presentation: An 86-year-old man complained of blurred vision and decreased visual acuity in his right eye 8 days after receiving the second BNT162b2 mRNA-based COVID-19 vaccine and was referred to our hospital. He also had pain with eye movement. Best corrected visual acuity (BCVA) in the right eye was 20/200 and critical flicker frequency dropped to 16 Hz. Relative afferent pupillary defect was positive and central scotomas were observed on visual field analysis. Fundus examination and SD-OCT revealed optic disc swelling and apparent thickening of the retinal nerve fiber layer around the optic disc in the right eye. Although either an increase in CRP or ESR on laboratory tests, demyelinating lesion on MRI, or positive of anti-MOG antibodies or anti-AQP4 antibodies were not observed, fluorescein angiography presented only hyperfluorescence of the optic disc in the right eye, but there were no findings such as papillary deficiency and choroidal delay that would suggest ischemic optic neuropathy. We diagnosed atypical optic neuritis developed after the SARS-CoV-2 mRNA-based vaccination and initiated oral corticosteroid therapy. One month later, the optic disc swelling disappeared and BCVA improved to 20/100; however, the central scotoma remained and no further improvement in visual function OD was obtained. Conclusions: An atypical acute idiopathic optic neuritis can occur after receiving the second vaccination with BNT162b2, which may present a limited response to corticosteroid therapy.

## 1. Background

Coronavirus disease 2019 (COVID-19) remains rampant worldwide, with the development of vaccines against the causative virus severe acute respiratory syndrome coronavirus 2 (SARS-CoV-2) progressing rapidly with an urgent demand. Several vaccines have been approved for emergency use for the prevention of COVID-19, though critical adverse events of the vaccines have not been fully investigated.

Multiple sclerosis (MS) or neuromyelitis optica spectrum disorders (NMOSD) after COVID-19 vaccination have been reported [1,2,3,4,5,6], while a complication of optic neuritis (ON) is also known [7,8,9,10]. Although idiopathic ON after COVID-19 vaccination is also reported [11,12,13,14,15], those visual prognoses are generally favorable. We report a case of atypical unilateral ON after receiving the BNT162b2 mRNA-based COVID-19 vaccine, in which the recovery of visual functions was restricted despite corticosteroid therapy.

## 2. Case Presentation

An 86-year-old man who complained of blurred vision and decreased visual acuity in the right eye visited a local eye clinic. This patient had received the second dose of the BNT162b2 mRNA-based COVID-19 vaccine 8 days before the onset of his ocular symptoms. Since the patient was also aware of pain with eye movement, he was referred to our hospital on suspicion of optic neuritis. He had a history of arrhythmia, but had no abnormalities such as flu-like symptoms before vaccination. At presentation, the best-corrected visual acuity (BCVA) was 20/200 in the right eye (OD) and 20/20 in the left eye (OS), with intraocular pressure (IOP) 14 mmHg OD and 15 mmHg OS. Critical flicker frequency (CFF) was 16 Hz OD and 47 Hz OS. Relative afferent pupillary defect (RAPD) OD was observed and fundus examination revealed optic disc swelling OD (Figure 1A,B). Fluorescein angiography (FA) OD showed only hyperfluorescence of the optic disc (Figure 1C,D), with no findings such as papillary filling deficiency and choroidal delay that would suggest ischemic optic neuropathy. Humphrey’s visual field analysis (HFA) and spectral domain optical coherence tomography (SD-OCT) OD revealed central scotomas (Figure 1E,F) and apparent thickening of the retinal nerve fiber layer (RNFL) around the optic disc (Figure 2). Alternatively, laboratory tests including erythrocyte sedimentation rate (ESR) and C-reactive protein (CRP) indicated no abnormal values. Anti-myelin oligodendrocyte glycoprotein (MOG) antibodies or anti-aquaporin 4 (AQP4) antibodies were also examined by commercial-based cell-based assays (CBAs) using live transfected cells, but were negative. Intracranial magnetic resonance imaging (MRI) indicated neither a contrast-enhanced effect of gadolinium on the optic nerve nor abnormalities such as demyelinating lesions. Non-arteritic anterior ischemic optic neuropathy (NA-AION) was suspected based on the patient’s age and his medical history of arrhythmia; however, FA findings and HFA results were consistent with acute idiopathic optic neuritis OD, contrary to NA-AION. Since the patient had received the second BNT162b2 mRNA-based COVID-19 vaccine 8 days before the onset of ocular symptoms OD, a side effect of the COVID-19 vaccine was suspected to be the possible cause. Considering the patient’s advanced age, methylprednisolone pulse administration was avoided and oral corticosteroid therapy was initiated from 0.6 mg/kg. One month later, although the optic disc swelling OD had gradually resolved (Figure 3A,B) and BCVA OD had improved to 20/100, the central scotoma remained (Figure 3C,D) and no further improvement in visual function OD was obtained. There was no inflammation in the left eye during these events.

## 3. Discussion

Optic neuropathy is suspected as one of adverse events of the COVID-19 vaccine, which includes ON associated with or without demyelinating CNS diseases [7,8,9,10,11,12,13,14,15,16], NA-AION [17,18,19,20,21,22,23,24], and NMOSD [8,9]. Lotan et al. reviewed 14 case reports and 2 case series by electronic searches of the published literature regarding neuro-ophthalmological complications of COVID-19 vaccines and reported that optic neuritis was the most common, occurring in 61 of 76 cases (80.3%) [25]. We reviewed the previous reports of newly onset optic neuropathy after receiving COVID-19 vaccines and compared clinical manifestations, management, and outcomes in Table 1. The onset after receiving COVID-19 vaccines ranged from 1 day to 3 weeks, in which 7 of all 26 cases (26.9%) were associated with relapsing-remitting multiple sclerosis (RRMS) or NMOSD, with 10 cases (38.5%) idiopathic ON. High-dose pulse steroid therapy was useful in most cases and their visual prognosis was favorable. Although visual acuity was not fully recovered in 3 cases [13], those were not as high as in this case. NA-AION was also reported in 8 cases (30.8%), with ON suspected in some cases.

Clinically, it is often challenging to differentiate ON from NA-AION, with the diagnosis usually based on the patient’s background, clinical findings, and multimodal images. In this case, NA-AION OD was primarily suspected based on the medical history of arrhythmia, the patient’s age, negative result for anti-MOG or anti-AQP4 antibodies, no elevation of ESR or CRP, and no associated neurological signs or abnormalities on an intracranial MRI. However, FA presented only hyperfluorescence of the optic disc, without papillary filling deficiency or choroidal delay suggesting AION was negative. In addition, visual field test results provide the critical clue of the diagnosis. It is well known that central scotomas are highly characteristic of ON, while an inferior altitudinal defect along the horizontal meridian, particularly in the nasal periphery, is characteristic of AION [26]. Although a worse visual outcome in this case is also differentiating features of NA-AION, eye movement pain, FA findings, and central scotomas without the horizontal meridian led us to diagnose acute ON rather than NA-AION. 

Hypotheses indicating a causal relationship between COVID-19 vaccination and ON are yet to be proved, but the close temporal association between symptom onset and vaccination strongly supports that possibility. BNT162b2 is a nucleoside-modified mRNA vaccine, which is translated into the SARS-CoV-2 spike protein by the host’s ribosomes, followed by antigen processing and presentation to local immune cells for subsequent neutralizing antibody production and T-cell-mediated immune response [27]. Since there have also been several reports regarding optic neuritis developed after SARS-CoV-2 infection, we speculate that adverse events after receiving the COVID-19 vaccine are probably attributed to the adaptive immune response evoked by the vaccination, and/or the spike protein itself. 

The autoimmune mechanism evoked by molecular mimicry of viral proteins and the immunological involvement of adjuvants have been suggested to underlie the development of ON following COVID-19 vaccination [7,28]. Visual prognosis of ON associated with RRMS or idiopathic ON with autoimmune nature is generally favorable by high-dose pulse steroid therapy. The reasons for the inadequate recovery of visual function in this case are speculated to be the avoidance of methylprednisolone pulse administration due to the patient’s advanced age and the impairment of tissue repair associated with aging.

In conclusion, we encountered a case of unilateral atypical ON occurring after the second BNT162b2 mRNA-based COVID-19 vaccination. Warnings should be given to ophthalmologists and physicians about the risk of atypical optic neuritis after COVID-19 vaccination.

## Figures and Tables

**Figure 1 vaccines-10-01574-f001:**
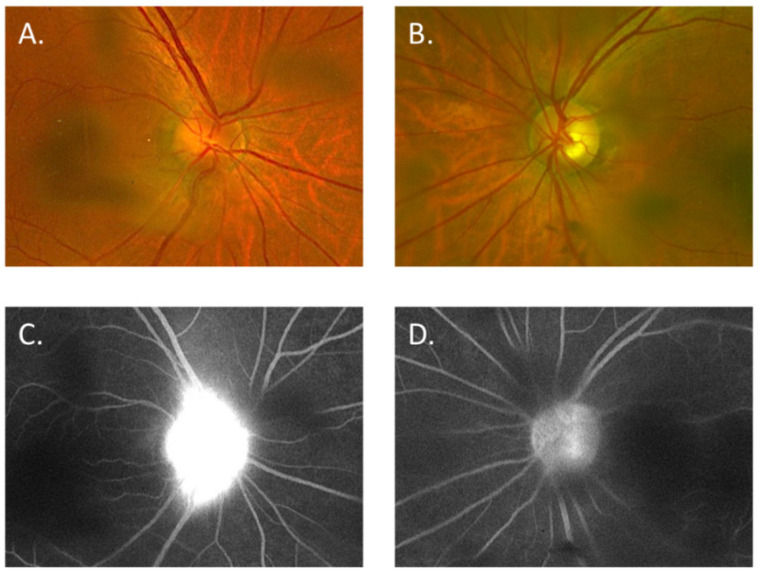
(**A**,**B**) color fundus photographs of the right eye (**A**) and left eye (**B**) 8 days after receiving the second COVID-19 vaccination. (**C**,**D**) fluorescein angiography findings in the right eye (**C**) and left eye (**D**) at the late stage. (**E**,**F**) Humphrey’s visual field analysis of the right eye (**E**) and left eye (**F**).

**Figure 2 vaccines-10-01574-f002:**
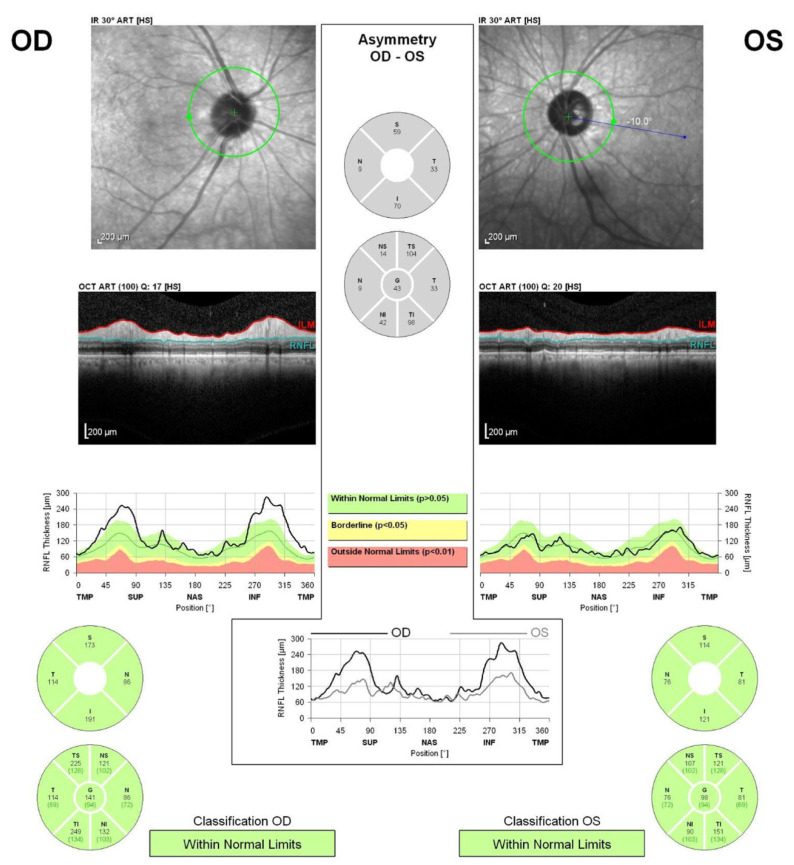
Retinal nerve fiber layer thickness around the optic nerve head of the right eye (**OD**) and left eye (**OS**) measured by spectral domain optical coherence tomography.

**Figure 3 vaccines-10-01574-f003:**
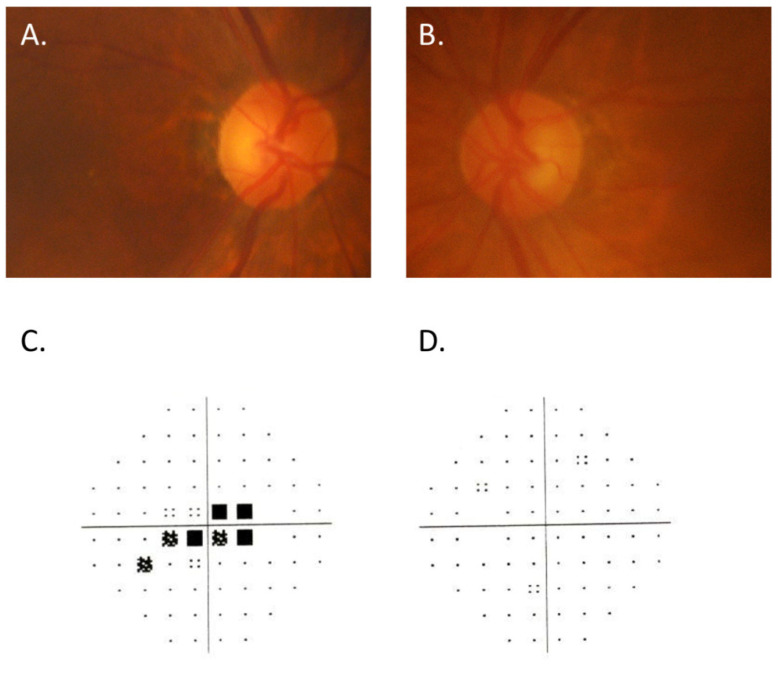
(**A**,**B**) color fundus photographs of the right eye (**A**) and left eye (**B**) 1 month after receiving the second COVID-19 vaccination. (**C**,**D**) Humphrey’s visual field analysis of the right eye (**C**) and left eye (**D**).

**Table 1 vaccines-10-01574-t001:** Published cases reports of optic neuropathy after receiving COVID-19 vaccines.

	Author (Year)	Age (Sex)	Medical History	Clinical Presentation	Diagnosis	Vaccine	Time between Vaccination and Symptoms	Treatment	Final Visual Acuity
1	Ayman G Elnahry (2021) [11]	69 (female)	None	Bilateral vision loss	Bilateral optic neuritis	mRNA (Pfizer-BioNTech) 2nd dose	16 days	1000 mg/day intravenous methylprednisolone (IVMP) for 5 days,	Stable
		32 (female)	None	Blurred vision OS	Optic neuritis OS	Viral vector (Oxford–AstraZeneca) 1st dose	5 days	1000 mg/day IVMP for 3 days	20/20
2	Henrique M. Leber (2021) [16]	32 (female)	None	Vision loss and pain OS	Bilateral optic neuritis and thyroiditis	Inactivated (Sinovac) 2nd dose	12 h	1000 mg/day IVMP for 5 days	20/20 OD 20/25 OS
3	Rika Tsukii (2021) [17]	55 (female)	None	Visual field disturbance OD	Non-arteritic anterior ischemic optic neuropathy (NA-AION) OD	mRNA (Pfizer-BioNTech) 1st dose	7 days	No treatment	20/20
4	Valentina Arnao (2022) [14]	Middle-age (female)	None	Bilateral blurred vision and pain	Bilateral optic neuritis	Viral vector (Oxford–AstraZeneca) 1st dose	14 days	1000 mg/day IVMP for 5 days	Recovered
5	Jiajun Wang (2022) [12]	21 (female)	None	Blurred vision OD with ocular rotation pain	Optic neuritis OD	Inactivated (Sinopharm) 2nd dose	3 weeks	800 mg/day IVMP for 3 days	20/20
		38 (female)	None	Blurred vision OD	Optic neuritis OD	Inactivated (Sinopharm) 1st dose	3 weeks	1000 mg/day IVMP for 3 days	20/20
6	Madhurima Roy (2022) [13]	27 (female)	None	Progressive blurring of vision OS	Optic neuritis OS	Viral vector (Covishield) 1st dose	9 days	1000 mg/day IVMP for 3 days	20/40
		48 (female)	None	Painless gradual diminution of vision OS	Optic neuritis OS	Viral vector (Covishield) 2nd dose	5 days	1000 mg/day IVMP for 3 days	20/30
		40 (male)	None	Blurring vision in both eyes	Bilateral optic neuritis	Viral vector (Covishield) 1st dose	12 days	Steroid therapy	20/30 OD 20/40 OS
7	Mahsa Khayat-Khoei (2022) [7]	26 (female)	None	Blurred vision and pain OD	New onset relapsing-remitting multiple sclerosis (RRMS)	mRNA (Moderna) 2nd dose	14 days	1000 mg/day IVMP for 5 days	-
		64 (male)	RRMS	Vision changes and pain OD	Multiple sclerosis (MS) exacerbation	mRNA (Pfizer-BioNTech) 2nd dose	1 day	1000 mg/day IVMP for 3 days	-
		33 (male)	none	Blurred vision OS	New onset RRMS	mRNA (Pfizer-BioNTech) 2nd dose	1 day	1000 mg/day IVMP for 3 days	-
		48 (female)	clinically isolated demyelinating syndrome (CIS)	Pain OD, Lhermitte’s, balance/gait	Conversion from CIS to RRMS	mRNA (Pfizer-BioNTech) 1st dose	15 days	1000 mg/day IVMP for 3 days	-
8	Christian García-Estrada (2022) [15]	19 (female)	None	Vision loss and pain OS	Optic neuritis OS	Viral vector (Janssen) 1st dose	1 week	1000 mg/day IVMP for 5 days	20/20
9	Yelda Yıldız Tascı (2022) [8]	32 (male)	Graves’ disease	Ocular pain and blurred vision OD	Neuromyelitis optica spectrum disorders (NMOSD) OD	Inactivated (Sinovac) 1st dose	14 days	1000 mg/day IVMP for 5 days	20/20
10	Sai A Nagaratnam (2022) [10]	36 (female)	None	Bilateral vision loss and subjective color desaturation, painful eye movements and fatigue	Bilateral optic neuritis	Viral vector (Oxford–AstraZeneca ChAdOx1) 1st dose	14 days	1000 mg/day IVMP for 3 days	20/16 OD 20/20 OS
11	Bader Shirah (2022) [9]	31 (female)	Systemic lupus erythematosus (SLE)	Painful eye movements and blurred vision OS	NMOSD OS	mRNA (Pfizer-BioNTech) 2nd dose	14 days	1000 mg/day IVMP for 2 days	-
12	Wen-Yun Lin (2022) [18]	61 (female)	Hypertension and hyperlipidemia	Blurred vision OS	NA-AION OS	Viral vector (Oxford–AstraZeneca ChAdOx1) 2nd dose	7 days	Oral prednisolone 60 mg/day	20/80
13	Abdelrahman M Elhusseiny (2022) [19]	51 (male)	None	Vision loss OS	NA-AION OS	mRNA (Pfizer-BioNTech) 2nd dose	1 day	Oral prednisone over 1 month	20/400
14	Sonia Valsero Franco (2022) [20]	53 (male)	None	Bilateral vision loss	Suspected bilateral NA-AION	mRNA (Pfizer-BioNTech) 1st dose OD mRNA (Pfizer-BioNTech) 2nd dose OS	7 days 10 days	Acetazolamide 750 mg/day	20/20 OD 20/40 OS
		65 (male)	Arterial hypertension	Blurred vision OD	Suspected NA-AION OD	mRNA (Pfizer-BioNTech) 1st dose	12 days	No treatment	20/200
15	Snezhana Murgova (2022) [21]	45 (male)	Arterial hypertension	Visual disturbance OD	NA-AION OD	mRNA (Pfizer-BioNTech) 2nd dose	10 days	Vasodilators and anti-platelet therapy	20/20
16	Seung Ah Chung (2022) [22]	65 (female)	None	Sudden inferior visual field loss OD	NA-AION OD	Viral vector (Oxford–AstraZeneca ChAdOx1) 2nd dose	15 days	1000 mg/day IVMP for 3 days	20/200
17	Ilay Caliskan (2022) [23]	43 (female)	None	Blurred vision and movement-associated pain OD	NMOSD OD	mRNA (Pfizer-BioNTech) 2nd dose	1 day	IVMP and plasma exchange	-
18	Srinivasan Sanjay (2022) [24]	50s (Female)	Non-arteritic anterior ischaemic optic neuropathy (NA-AION) OD	Vision loss OS	NA-AION OS	Viral vector (Covishield) 1st dose	4 days	Oral aspirin 75 mg for 1 month	20/20

## Data Availability

Not applicable.

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
