# Peer review of "A Case of Atypical Unilateral Optic Neuritis Following BNT162b2 mRNA COVID-19 Vaccination"

_vaccines, 2022, doi:10.3390/vaccines10101574_

Round 1
Reviewer 1 Report
This paper reported reported a case of an atypical acute idiopathic optic neuritis which was occurred after receiving the second vaccination with BNT162b2. Up today, multiple ocular diseases has been observed after COVID-19 vaccines. This report showed detailed information and the observation was consistant with the previous discoveries. Overall, it is a pleasant paper to read. The result is of interests to the vaccine community.
I have two minor concerns.
1) The authors suspected the adverse event is probably attributed to the adaptive immune response evoked by the vaccination, and/or the spike protein itself. I was wondering whether any similar adverse events observed in the SARS-Cov2 infected patients?
2) The adverse event presents a limited response to corticosteroid therapy. What is the reason, if it the results of the adaptive immune response evoked by the vaccination. Please provide more discussion.
Author Response
Reviewer #1
We thank the reviewer for the careful comments that contained important suggestions to improve our manuscript. We have revised the manuscript according to their instructions. Below is a list of point-to-point responses to their comments.
1) The authors suspected the adverse event is probably attributed to the adaptive immune response evoked by the vaccination, and/or the spike protein itself. I was wondering whether any similar adverse events observed in the SARS-Cov2 infected patients?
Response: Since there are also several reports regarding optic neuritis developed after SARS-CoV-2 infection, we speculate that adverse events after receiving COVID-19 vaccine are probably attributed to the adaptive immune response evoked by the vaccination, and/or the spike protein itself. We revised the corresponding paragraph.
2) The adverse event presents a limited response to corticosteroid therapy. What is the reason, if it the results of the adaptive immune response evoked by the vaccination. Please provide more discussion.
Response: We appreciate the reviewer’s advice. Visual prognosis of ON associated with RRMS or idiopathic ON with autoimmune nature is generally favorable by high-dose pulse steroid therapy. The reasons for the inadequate recovery of visual function in this case are speculated to be the avoidance of methylprednisolone pulse administration due to the patient's advanced age and the impairment of tissue repair associated with aging. We added this description in the Discussion.
Reviewer 2 Report
The paper is clear and well written and present a well documented review of the possible association between COVID19 vaccination (any type) and atypical optic neuritis. I have only few points to improve and some suggestions:
1. please review line 97 (.....reported in observed in.....)
2. it is important to specify the method employed to test anti-MOG and anti-AQP4 antibodies since commercial methods with fixed transfected cells may be slightly less sensitive than methods employing live transfected cells.
3. the authors declare that CRP and ESR were normal; did they have the possibility to test other inflammatory markers? It would be of great interest to test for example IL-6, IL-8, TNF, IL-17 and sIL-2R, not only in serum but also in CSF. I know that these markers are not routinary tested in the lab, but in some cases may be useful to investigate atypical immune-mediated disease like this ON, where common inflammatory markers are usually negative. If not able to do retrospectively, please add a comment on this.
4. Did the patient have COVID19 infection or other flu-like manifestation before vaccination? This is an important information. And would be of great importance also for all the other reported cases from the literature, if possible. Especially in cases occurred after the first dose of vaccine.
5. A final comment. In our Laboratory when assessing MOG and AQP4 antibodies we always add an indirect immunoflorescence test on cerebellar tissue to test the possible presence of any type of neurological Ab or reveal a new possible reactivity versus neuronal tissue. Did or do you have the possibility to test this in your patient? Since the optic nerve was involved it would be of great interest to test patient serum against optic nerve tissue and maybe to perform the same test adding increasing concentration of spike protein in order to evaluate a possible implication of anti-spike antibodies in the disorder.
Author Response
Reviewer #2
We thank the reviewer for the careful comments that contained important suggestions to improve our manuscript. We have revised the manuscript according to their instructions. Below is a list of point-to-point responses to their comments.
- please review line 97 (.....reported in observed in.....)
Response: We apologize for our writing mistake. We removed “observed in” from the sentence.
- it is important to specify the method employed to test anti-MOG and anti-AQP4 antibodies since commercial methods with fixed transfected cells may be slightly less sensitive than methods employing live transfected cells.
Response: Both anti-MOG and anti-AQP4 antibodies were examined by commercial based cell based assays (CBAs) used live transfected cells. We revised the corresponding sentence as follows.
“Anti-myelin oligodendrocyte glycoprotein (MOG) antibodies or anti-aquaporin 4 (AQP4) antibodies were also examined by commercial based cell based assays (CBAs) used live transfected cells, but were negative.”
- the authors declare that CRP and ESR were normal; did they have the possibility to test other inflammatory markers? It would be of great interest to test for example IL-6, IL-8, TNF, IL-17 and sIL-2R, not only in serum but also in CSF. I know that these markers are not routinary tested in the lab, but in some cases may be useful to investigate atypical immune-mediated disease like this ON, where common inflammatory markers are usually negative. If not able to do retrospectively, please add a comment on this.
Response: We appreciate the reviewer's constructive suggestion. However, since the prior clinic is now following up this patient, we are sorry that we won't see him unless his condition changes.
- Did the patient have COVID19 infection or other flu-like manifestation before vaccination? This is an important information. And would be of great importance also for all the other reported cases from the literature, if possible. Especially in cases occurred after the first dose of vaccine.
Response: He had a history of arrhythmia, however had no abnormalities such as flu-like symptoms before this vaccination. We have revised the corresponding sentence as follows.
“He had a history of arrhythmia, however had no abnormalities such as flu-like symptoms before this vaccination.”
- A final comment. In our Laboratory when assessing MOG and AQP4 antibodies we always add an indirect immunoflorescence test on cerebellar tissue to test the possible presence of any type of neurological Ab or reveal a new possible reactivity versus neuronal tissue. Did or do you have the possibility to test this in your patient? Since the optic nerve was involved it would be of great interest to test patient serum against optic nerve tissue and maybe to perform the same test adding increasing concentration of spike protein in order to evaluate a possible implication of anti-spike antibodies in the disorder.
Response: We appreciate for your valuable advice. Although we have not done, yet, it may provide us useful information in the diagnosis and treatment of idiopathic ON. We will make it a reference.